# Recent Advances in Pancreatic Ductal Adenocarcinoma: Strategies to Optimise the Perioperative Nutritional Status in Pancreatoduodenectomy Patients

**DOI:** 10.3390/cancers15092466

**Published:** 2023-04-25

**Authors:** James M. Halle-Smith, Sarah F. Powell-Brett, Lewis A. Hall, Sinead N. Duggan, Oonagh Griffin, Mary E. Phillips, Keith J. Roberts

**Affiliations:** 1Hepatobiliary and Pancreatic Surgery Unit, Queen Elizabeth Hospital Birmingham, Birmingham B15 2TH, UK; 2College of Medical and Dental Sciences, University of Birmingham, Birmingham B15 2TH, UK; 3Department of Surgery, Trinity College Dublin, University of Dublin, Tallaght University Hospital, D24 NR0A Dublin, Ireland; 4Department of Nutrition and Dietetics, St. Vincent’s University Hospital, D04 T6F4 Dublin, Ireland; 5Department of Nutrition and Dietetics, Royal Surrey County Hospital, Guildford GU2 7XX, UK

**Keywords:** pancreatic ductal adenocarcinoma, surgery, nutrition

## Abstract

**Simple Summary:**

Pancreatic cancer is an aggressive cancer, and the surgery to remove it carries significant risks. If patients undergo successful surgery, then they will require chemotherapy to stop the cancer from coming back. Poor nutrition increases the risks of surgery and reduces the number of patients that can have chemotherapy. In this article, we describe various strategies that can improve the nutrition of these patients before, during, and after surgery so as to lead to the best treatment outcomes. We also describe areas in this field that will benefit from future research so that we can continue to improve the treatment outcomes for this vulnerable patient group.

**Abstract:**

Pancreatic ductal adenocarcinoma (PDAC) is an aggressive malignancy for which the mainstay of treatment is surgical resection, followed by adjuvant chemotherapy. Patients with PDAC are disproportionately affected by malnutrition, which increases the rate of perioperative morbidity and mortality, as well as reducing the chance of completing adjuvant chemotherapy. This review presents the current evidence for pre-, intra-, and post-operative strategies to improve the nutritional status of PDAC patients. Such preoperative strategies include accurate assessment of nutritional status, diagnosis and appropriate treatment of pancreatic exocrine insufficiency, and prehabilitation. Postoperative interventions include accurate monitoring of nutritional intake and proactive use of supplementary feeding methods, as required. There is early evidence to suggest that perioperative supplementation with immunonutrition and probiotics may be beneficial, but further study and understanding of the underlying mechanism of action are required.

## 1. Introduction

Realising good outcomes in pancreatic surgery is challenging, but achievable [1,2]. Poor prognosis tumours and the high rates of perioperative morbidity and mortality associated with resection tend to dominate the discussion around improving outcomes [2,3]. Evolution of systemic therapies [4,5] and organisational changes, such as centralised surgery [6], are evidence of this, but they require large investment of time and finance. 

Malnutrition and maldigestion are highly prevalent in this patient population and contribute to poor outcomes. However, despite clear evidence of suboptimal care and unmet need, there is far less focus on these issues [7,8,9,10]. Perioperative malnutrition has been identified by the American College of Surgeons National Surgical Quality Improvement Program (NSQIP) as one of the major modifiable risk factors associated with poor surgical outcomes, including mortality after surgery [11,12]. Malnutrition is present in a fifth of patients before pancreatoduodenectomy (PD) [9] and increases during inpatient stay, to greater than 75% [8]. The underlying cause is multifactorial, with pancreatic exocrine insufficiency (PEI) playing a major role. Figure 1 summarises the causes of malnutrition and maldigestion in this patient population. 

Consequences of untreated malnutrition and maldigestion are far reaching, contributing to reduced quality of life, decreased physical functioning, and weight loss in the short term, increasing rates of perioperative complications and jeopardising the chances of patients receiving anticancer treatments [13] in the medium term. Additionally, in the long term, they contribute to poorer quality of life, nutrient deficiency, and reduced survival [8,9,14]. 

The benefits of pancreatic enzyme replacement therapy (PERT) are clear [15], but there is a lack of consensus as to the best way to provide nutritional support to PD patients postoperatively. The European Society for Clinical Nutrition and Metabolism (ESPEN) guidelines for nutritional management after surgery recommend nutritional intervention if patients are not meeting certain requirements [16]. Many PD patients struggle to meet these requirements due to postoperative pain, delayed gastric emptying (DGE), and PEI [8,17]. The most recent enhanced recovery after surgery (ERAS) guidelines does give some guide as to nutritional intervention, but they acknowledge that the best route to deliver this nutrition is not yet known, and the recommendations they have made are predominantly based on derived data from other surgical cohorts [18]. Strategies that could improve perioperative nutritional management of these patients and reduce malnutrition rates would, therefore, have a wide range of benefits to both patients and health services alike.

In summary, to improve pancreatic surgery outcomes, every step of the care pathway should be optimised. Malnutrition and maldigestion are rife in this patient group, and often poorly managed. The aims of this review are, therefore, to summarise the current evidence for perioperative nutritional interventions in PD patients, and then the next step is to identify strategies that may warrant further investigation.

## 2. Preoperative Strategies

### 2.1. Preoperative Nutritional Assessment

Between 20–30% of patients are already malnourished before undergoing PD [8,9], which is associated with increased postoperative morbidity [9]. There are many factors specific to PD patients that contribute to the increased preoperative malnutrition rates amongst this group, and these include PEI, gastric outlet obstruction (GOO), and cancer cachexia, which is present in a large proportion of pancreatic cancer patients at diagnosis [19,20,21] (Figure 1). As such, preoperative assessment by a specialist dietitian is a useful way to identify these patients and intervene as required [22]. Indeed, the European Society for Clinical Nutrition and Metabolism (ESPEN) consensus guidelines for nutrition in surgical patients recommend nutritional status to be assessed before and after major surgery, such as PD, using the Nutritional Risk Screening (NRS-2002) [16]. Other methods include bioimpedance vector analysis (BIVA), which is a tool used to evaluate body composition (adipose and lean tissue, as well as muscle and visceral fat tissue) and hydration status (total, intra-, and extra-cellular water). BIVA has been used to assess nutritional status in cancer patients, and there is good evidence that this can be used to predict short- and long-term outcomes. Specific to pancreatic cancer, BIVA has been shown to predict major morbidity following resection, but further evidence is needed to define its role in routine pre-operative assessment [23,24,25]. For those identified as having significant preoperative malnutrition, individualised preoperative optimisation should be considered, with multidisciplinary input that could involve the use of preoperative nasojejunal (NJ) feeding.

Assessment of nutritional assessment is complex, and it is now widely accepted that assessment of muscle function, strength, and mass is required to allow timely identification of sarcopenia across all categories of BMI [26,27,28]. Simplified nutritional screening tools result in a wide range of results and are not correlated with surgical outcomes [29]. 

Given that preoperative malnutrition is a significant independent predictor of overall complications following PD [9], timely screening would allow identification of those patients at greatest risk and could enable preoperative optimisation of this modifiable risk factor. Nutritional assessment would help clinicians and patients make well informed, personalised decisions regarding nutritional supplements and support. Clearly, the window for this may be small given the rise in ‘fast-track’ resections for periampullary malignancies [30], but this should be a target nevertheless. 

### 2.2. Prehabilitation

Early synthesised evidence suggests that prehabilitation may improve outcomes after pancreatic surgery but, at present, there is a lack of standardisation with programmes varying in contents and length [31,32]. Specific to nutrition, there have been reports that prehabilitation programmes can slow the nutritional deterioration amongst the patient group and, in doing so, reduce rates of preoperative malnutrition [33]. 

### 2.3. Diagnosis and Management of Preoperative Pancreatic Exocrine Insufficiency

Patients with an unresected head of pancreas tumour are at risk of PEI, both from lower enzyme secretion (tumour burden) and from reduced enzyme delivery and activation owing to pancreatic duct obstruction [34]. The incidence of PEI pre-operatively is challenging to assess, and no ideal diagnostic test exists. The secretin test and coefficient of fat absorption (CFA) are accurate, but they are expensive and unpleasant, whilst faecal elastase (FE-1), although suitable for routine use, has poor sensitivity in mild PEI [35]. A systematic review in 2016 identified four studies evaluating PEI pre-operatively, all using FE-1, and found a median prevalence of 44% (range 42–47%) [36]. A new diagnosis of resectable pancreatic cancer is a challenging time for patients with rapid pre-operative work up and a race to theatre, and consequently their nutritional status is often overlooked. However, the impact of PEI on quality of life and patient outcomes should not be underestimated (which is discussed in full in the post-operative section) and should be considered as organ failure. PERT should, therefore, be considered for all with a pancreatic head malignancy awaiting resection. This is reflected in the UK NICE guidelines (National Institute for Clinical Excellence [37]). 

### 2.4. Other Preoperative Considerations

As highlighted in Figure 1, there are many factors that contribute to malnutrition and maldigestion in a patient with unresected pancreatic cancer. It is important to balance delays to operative intervention and the need for patient optimisation. Jaundice is highly prevalent among this patient population. Jaundice exacerbates maldigestion and is frequently treated by biliary drainage. However, this takes time, and resolution of jaundice is relatively slow [30]. A surgery-first approach has benefits of reducing perioperative complication rates, but it also definitely treats the cause of obstructive jaundice [38]. Due the need for urgent surgery in this patient population, jaundice is thus addressed quickly in the patient pathway, and, further, the drop in bilirubin is more rapid [39]. Therefore, a direct-to-surgery approach for jaundiced patients with resectable pancreatic cancer is preferable, whenever possible, to biliary drainage [1]. Gastric outlet obstruction (GOO) is less common than jaundice, but it is potentially more harmful. Early surgery is preferable when feasible, with duodenal stents reserved for palliative treatment [40]. Correcting electrolyte disturbance is essential, and the debilitating effects of cancer cachexia must be kept in mind, but the mechanisms are complex and beyond scope of this review [19,20].

## 3. Intraoperative Strategies

On the day of surgery, ESPEN guidelines recommend that clear fluids may be permitted up to two hours before induction of anaesthesia and solid foods up to six hours before induction [16,41]. That said, it should be acknowledged that this may be hazardous in the pancreatic cancer population who may have some degree of GOO [16,41], so individual patient assessment is advisable. 

Similarly, it has been shown that carbohydrate loading drinks can reduce the harmful glycogen-depleted state induced by overnight fasting and have been shown also to reduce postoperative insulin resistance, thirst, and anxiety in patients undergoing elective surgery [41,42,43,44]. Whilst these findings are not specific to pancreatic surgery patients, the beneficial effects of this simple intervention would be valuable in this population known to be at significant risk of perioperative malnutrition and dehydration. However, due to the risk of pre-operative new onset diabetes due to pancreatic endocrine insufficiency, all patients should be screened for diabetes prior to the administration of carbohydrate loading drinks to prevent significant hyperglycaemia in the immediate pre-operative period. 

Intraoperatively, consideration of postoperative nutritional routes should also be made. For example, despite evidence suggesting that those with preoperative GOO can safely take oral diet after PD [45], it may be wise to keep one central line port for postoperative parenteral nutrition (PN). Alternatively, an intra-operative NJ feeding tube could be placed, depending on individual patient assessment. Furthermore, whilst current evidence suggests that oral feeding in post-operative pancreatic fistula (POPF) does not prolong healing or increase the chance of associated complications such as post-pancreatectomy haemorrhage (PPH) [46], patients with CR-POPF are known to have greater risk of ileus and DGE. Therefore, those who are thought to be at a high risk of POPF, according to widely known risk calculators [47,48], may also benefit from an alternative feeding route, such as NJ or PN, which can be placed during the index operation [49]. Drainage naso-gastric (NG) tubes should not be routinely placed, as a recent systematic review and meta-analysis suggests that routine NG drainage was associated with increased rates of DGE and length of stay after PD [50]. Whilst not specifically related to nutrition, it is important to acknowledge the role of intraoperative fluid management on the outcome of PD patients. Higher volume of intraoperative blood loss and greater number of units transfused is associated with poorer outcomes after PD [51,52,53]. Therefore, strategies to reduce intraoperative blood loss and transfusion requirement are desirable, and examples of these include tranexamic acid (TXA) and autologous blood transfusion via recovery devices. Regarding TXA, assimilated evidence shows that a dose of 1 g for adults can reduce surgical blood loss by around 30% without increasing risk of adverse events, but robust evidence specific to PD is still awaited [54]. Autologous transfusion has historically been avoided in cancer surgery due to fear of disseminating circulating tumour cells (CTC). However, recent evidence suggests this may be safer than initially thought amongst PDAC patients [55].

## 4. Postoperative Strategies

### 4.1. Monitoring Nutritional Intake after Pancreatoduodenectomy

The ESPEN consensus guidelines for nutritional management after surgery recommend nutritional intervention if patients are not meeting 50% of their nutritional requirements after surgery, which they estimate at 1.5 g/kg/day of protein and 25–30 kcal/kg/day [16]. Similarly, the most recent enhanced recovery after surgery (ERAS) guidelines for PD advise additional routes to deliver nutrition if patients are not meeting 60% [41] of their nutritional requirements for seven days after surgery. However, it remains difficult to accurately track nutritional intake after surgery, in part due to the inability to allocate sufficient staff resources. A previous study, published in 2009, investigated perioperative nutritional practice for PD patients in the United Kingdom and highlighted the lack of specialist hepatobiliary dietetic support for PD patients [56]. Dietary monitoring is time consuming for busy and under-resourced dietetic and nursing staff, and, additionally, it has been shown that food intake recorded by nursing staff may not be accurate [57]. Therefore, future studies in this area should focus on feasible methods for recording patient dietary intake. One potential method is via electronic food charts [58,59], enabling patients to take control of recording their own intake, and this allows clinical teams to monitor intake remotely and intervene appropriately. A recent systematic review also showed that patients were able to record their intake accurately via dietary record apps [60].

There is evidence that many patients struggle to meet these requirements, with recent publications reporting a typical intake of between 27–32 g of protein and 588–745 kcal per day [8,17]. This is likely due to a combination of factors, including postoperative pain, DGE, and PEI. However, there is a lack of provision for dedicated dietetics services to closely track the intake for these vulnerable patients, [56] which prevents prompt intervention when patients are falling behind their requirement. Even when the need for extra nutritional intervention is identified, the ERAS after PD guidelines acknowledge that the best route to deliver this nutrition is not yet known [18]. There are, however, some strategies that show promising early evidence, and these are described below.

### 4.2. Early Oral Feeding

Early oral feeding (EOF) is recommended by the most recent ERAS guidelines for care after PD, advising normal diet according to tolerance [41]. This is also in line with the broader ESPEN guidelines, which recommend early oral nutrition wherever safe after surgery [16]. However, there is no evidence to suggest that EOF is of benefit for longer term outcomes, such as the initiation of adjuvant chemotherapy, and whilst short term outcomes, such as length of hospital stay may be improved, the impact on overall recovery is unknown. There may also be a degree of clinician reluctance surrounding EOF after PD with concern over POPF or DGE risk. However, current evidence from a recent systematic review of observational studies suggests that EOF is safe after PD and does not increase POPF or DGE [61,62]. Physiologically, this seems logical, given that the main hormonal stimulator of pancreatic exocrine secretion, cholecystokinin (CCK), is produced in the duodenum [63,64,65], which is resected in PD. Furthermore, the majority of autonomic nerves, which stimulate the pancreas, are divided during a pancreatoduodenectomy when arteries, such as the gastroduodenal artery (GDA) [66], are ligated. Importantly, EOF has been shown to increase the calorie and protein intake in PD patients compared to standard care, increasing the average protein intake from 32–40 g/day and calorie intake from 745 kcal to 847 kcal/day per day [17]. However, it is important to recognise that the evidence for this remains limited to small observational studies, neither of these nutritional intakes are adequate, and further evidence of higher quality would be desirable to further investigate the effects of early oral nutrition in this group. 

Whilst current evidence would seem to suggest that EOF does not increase rates of POPF, any nutritional strategy after PD must be designed with the possibility of POPF in mind, given that it remains the main cause of morbidity after PD [67,68]. It is, therefore, important to note that current evidence from the literature suggests oral feeding in patients with POPF neither delays healing of the fistula nor increases the chances of associated complications, such as post-pancreatectomy haemorrhage (PPH) [46]. 

A further consideration for EOF after PD is that roughly 15% have preoperative symptoms of GOO. Some may be concerned that EOF in these patients could be unsafe. However, a recent retrospective study showed that there was no difference in DGE between patients fed orally compared to those fed with a NJ tube [45]. This study suggests that preoperative GOO should not be a contraindication to EOF after PD, but it may contribute to poor dietary intake, with the median time to meeting nutritional requirements being 12–14 days. An important finding of this study is that 49% of the patients in the oral feeding group required a feeding tube to be placed postoperatively [45]. NJ tubes are typically placed endoscopically, which anecdotally can be complicated after PD, and, for this reason, some centres place a NJ tube routinely at the time of surgery in case of need [30].

The presence of a NG tube for prolonged periods after surgery will also limit a patient’s ability to meet their nutritional needs orally, and the most recent ERAS after PD advise against routine NG decompression following PD [41]. Indeed, a recent systematic review and meta-analysis [50] investigated the impact of routine NG decompression after PD, and this showed an increase in DGE, as well as major complications and length of stay with routine NG decompression. 

### 4.3. Enteral Tube Feeding

That said, given that some patients are not to meet their nutritional requirements orally, for example, due to DGE, a supplementary feeding strategy is often required after PD. NJ feeding is frequently the favoured route, given that tubes can be placed intraoperatively, feeding can continue in patients with DGE, and it avoids the complications of the central venous access required for total parenteral nutrition [49,69,70]. Recent evidence, including a systematic review and meta-analysis [69,71], shows that EN is associated with fewer complications and reduced length of stay compared to PN after PD. Some reasons behind these clinical benefits are thought to be improved immunological function due to stimulation of enterocyte growth, stabilisation of the mucosal barrier, and a decrease in bacterial translocation [72,73,74]. In addition to this, cyclical NJ feeding, whereby feed is stopped for a period each day, has been shown to lead to faster recovery and progression to normal diet after PD compared to continuous feeding (RCT), likely due to more physiological cholecystokinin (CCK) levels [70]. Of note, other studies comparing EN to EOF have used continuous EN, so further research should include the use of cyclical NJ feeding [70]. However, one important complication to be aware of in postoperative enteral tube feeding is small bowel ischaemia. The exact mechanism for this is still not fully understood, but it is likely to involve large fluid shifts and osmotic changes [74,75,76]. To reduce the chances of this, the feeding rate should be increased slowly, and caution should be applied in patients who are hypotensive and requiring inotropic or vasopressor support [74,77]. Patients receiving small bowel feeding should be adequately hydrated prior to initiation of feeds, and EN should be discontinued if the patient becomes haemodynamically unstable. 

### 4.4. Diagnosis and Management of Postoperative Pancreatic Exocrine Insufficiency 

The aetiology of PEI, following PD, is more complex than just parenchymal loss from resection. One must also consider the anatomical and physiological changes resulting from reconstruction. Loss of the duodenum and creation of a pancreatico-jejunostomy or pancreatico-gastrostomy results in enzymes being delivered into an environment that is more acidic and lacking in enterokinase. An altered point of enzyme delivery, combined with denervation from resection, results in asynchrony between enzyme secretion and delivery of chyme. The above culminates in insufficient enzymes being delivered to the wrong place, at the wrong time, and at the wrong pH. 

The aetiology of PEI following resection also makes diagnosis challenging, PEI is far more than simply a lack of enzymes in this cohort. Therefore, direct tests of enzyme secretion will not reflect the true incidence. This may be why FE-1 significantly underestimates PEI following pancreatic resection [35]. Diagnosing PEI following pancreatic resection requires a test of enzyme function, rather than enzyme secretion. Current options include CFA or the ^13^C labelled mixed triglyceride breath test (^13^C-MTGT). CFA is considered the gold standard for fat malabsorption, but it is time-consuming and unpleasant for patients and laboratory staff alike. The ^13^C-MTGT is safe, non-invasive, and accurate. However, it requires a 6-h testing timeframe and is not widely available [78]. Symptomatic assessment is not reliable because the cardinal feature of steatorrhoea is present only with significant disease, and patients may unconsciously mask steatorrhoea by avoiding fatty food. The systematic review by Tseng et al reported a median PEI prevalence of 74% (range 36–100%) following PD, and the studies included mostly used FE-1 and are therefore likely to underestimate the true prevalence [36]. It is important to consider the progressive nature of PEI after PD, and studies that followed-up patients for longer report incidences approaching 100% [79,80]. The risk and incidence may increase where radiotherapy has been administered as treatment for pancreatic cancer, either prior to, or following resection [81,82,83,84,85].

Untreated PEI has a significant impact on quality of life with mentally and physically distressing symptoms including frequency, urgency, bloating, diarrhoea, fatty stool flatulence, loss of appetite, and vomiting [86,87]. The major clinical consequence of PEI is maldigestion, resulting in malabsorption, malnutrition, and the ensuing nutritional deficiencies (including albumin, pre-albumin, transferrin, lipoproteins, fat soluble vitamins, calcium, magnesium, zinc, thiamine, and folic acid [88,89]). PEI is associated with an increased risk of osteoporosis, sarcopenia, and cardiovascular events [90,91,92]. In pancreatic cancer, sarcopenia has been associated with increased perioperative mortality and reduced overall survival. PEI has also been shown to increase length of stay, post-operative complications, and costs following pancreatic resection [93,94].

PERT is the foundation of PEI management and has been shown to improve quality of life, mitigate weight loss, and even improve survival [15,95]. In an observational cohort study of 469 patients undergoing PD for cancer, patients on PERT survived 6.4 months longer than patients who were not on PERT, and the use of PERT was associated with improved survival in multivariate and propensity matched models [96]. Prospective trials specific to PEI following pancreatic resection are limited. A double-blind, placebo-controlled trial of 75,000 units of lipase per meal in patients with PEI following pancreatic resection showed significant improvements in nitrogen and fat absorption after only one week and significant symptom and weight improvements at one year. This is important, as it highlights the role of PERT in protein digestion, as well as fat. The UK NICE guidelines recommend consideration of PERT treatment for all following pancreatic resection [37,97] without the need for a positive diagnostic test, and this was supported by UK consensus guidelines released in 2021 [97]. Despite the high prevalence of PEI following PD, the significant effect it has on clinical outcomes, as well as the clear benefits of PERT treatment, it is still frequently undertreated [10]. 

The International Study Group on Pancreatic Surgery recommend starting with 40,000–50,000 units with main meals, and half of this should be represented by snacks [98]. Other studies recommend 75,000 units with main meals and 50,000 units with snacks [99,100]. The prospective trials described above suggest that the latter is more appropriate following pancreatic resection. The optimal timing of PERT consumption is spread throughout the meal, and this has been shown to reduce abdominal symptoms and improve weight gain [101,102]. The addition of a PPI is still an area of poor agreement and limited data, and recent UK consensus guidelines advised that PPI should be added as a second-line therapy if PERT alone is not effective [97]. However, following PPPD, routine use of a PPI is recommended to reduce the risk of ulceration at the gastrojejunostomy. Dose escalation and treatment monitoring are extremely important after PD and should be guided by a dietitian. There is no maximum dose in adults, but alternate or concurrent pathology (such as small intestinal bacterial overgrowth and bile acid malabsorption) should be considered where symptoms persist, despite doses exceeding 100,000 units per meal [103,104], or where dose escalation does not appear to result in a clinical benefit. An illustrated summary of interventions to improve nutritional status of PD patients is shown in Figure 2. 

## 5. Perioperative Supplementation

Various nutritional supplements, including immunonutrition (IM), probiotics, and synbiotics, have been investigated in PD patients and are thought to reduce infectious complications, a major source of morbidity after PD.

### 5.1. Immunonutrition

IM, which may contain arginine, omega-3 fatty acids, and/or ribonucleic acid, can be administered orally or via enteral tube in the preoperative or postoperative setting. It is hypothesised that IM can modulate immune responses, inflammatory pathways, and gut function after surgery. Indeed, a recent meta-analysis of RCTs, comparing IM to standard care after PD, showed a reduction in infectious complications amongst those who received IM [105]. However, this was only reported as a moderate GRADE recommendation, and there was no significant difference in major or PD-related complications [105]. The mechanisms behind a possible beneficial effect of IM in PD patients remain unclear. Some hypothesise that it may be due to a downregulation of prostaglandins and inflammatory cytokines, such as tumour necrosis factor-alpha (TNF-alpha) and interleukin-6 (IL-6) after PD [106,107,108]. However, this theory of hypocytokinaemia remains controversial [108,109]. Furthermore, patients receiving IM have been shown to have lower prostaglandin E2 levels, which has been associated with reduced postoperative complications [107]. Further investigation into the possible beneficial action of IM is therefore required. A further important consideration when administering IM in PD patients is that PERT should be administered concurrently, which can be logistically difficult.

### 5.2. Probiotics and Synbiotics

Probiotics are microorganisms thought to be beneficial to humans when supplemented in appropriate amounts [110], whilst prebiotics are substances (such as inulin) which promote the growth of beneficial gut microorganisms [111]. Synbiotics are a combination of pre- and probiotics [110]. A recent meta-analysis of RCTs [112] showed that probiotics and synbiotics can reduce infectious complications, length of stay, antibiotic use, and DGE in PD patients. Again, the mechanisms for this are not completely understood and require further investigation. However, it is thought that they may include promoting a healthier gut microbiome and reducing harmful bacterial translocation [113,114]. An important safety consideration is that there have been reported cases of *Lactobacillus* sepsis [115] and bowel ischaemia in patients with acute pancreatitis. However, in the latter study, a novel probiotic was administered into the small bowel in patients at high risk of poor bowel perfusion [116], and, therefore, its applicability is uncertain. 

## 6. Future Research Directions

This review describes multiple potential targets for improving patient outcomes through nutritional interventions before and after pancreatoduodenectomy. However, it is imperative that smaller work, suggestive of positive impact, is translated into robust, well designed trials that can generate high-level evidence and ultimately feed into clinical practice changes that benefit patients. Determining the direction of future research first requires evaluation of current variation in care and adherence to best practice guidelines, and this will determine the scope and potential need for nutritional interventions within and across geographical boundaries. For example, the UK national audit RICOCHET highlighted wide variation in practice and poor adherence to guidelines regarding PERT usage [10]. Future research should, therefore, continue to work towards improving the diagnosis of PEI (ideally with an acceptable test capable of evaluating PERT efficacy) and the delivery of PERT to patients with pancreatic cancer. 

Strategies to ensure that patients receive well evidenced nutrition, delivered in an appropriate fashion, before surgery, immediately after surgery, and in the longer term, need to be developed. Ideally, these will be patient led and goal directed to enable longer term benefit and minimise the requirement for high-cost resources. 

Designing clinical trials to robustly assess the efficacy of different nutritional therapies in pancreatic cancer surgery patients, such as immunonutrition and probiotics, has historically been fraught with difficulties. One aspect of this is the lack of an accepted ‘standardised practice’, which means that designing an appropriate control group to minimise the impact of important confounders is challenging. Additionally, pancreatic surgery trials typically struggle with timely recruitment and therefore are frequently underpowered to detect meaningful differences [117]. Clear, standardised outcomes to evaluate the effect of nutritional therapies have also been lacking, making designing and powering clinical trials extremely difficult. That said, there is reason for optimism given the recent publication of a core outcome set (COS) for nutritional and metabolic clinical effectiveness trials [118], as well as a planned trial which aims to evaluate the efficacy of preoperative nutritional therapy in Crohn’s disease (the OCEaN trial) being funded by the National Institute for Health Research (NIHR) [119]. The use of more pragmatic outcomes, such as the physical SF-36 scale and global leadership initiative on malnutrition (GLIM) criteria, which reflects the global effects of malnutrition [28], should be considered in future trial design.

## 7. Conclusions

In summary, optimising strategies to deliver best evidenced nutrition in a timely fashion to patients at all stages of their care pathway is imperative to improving patient outcomes. Defining current practice and variation in practice will enable the development of clinical trials to define optimum care and translate promising research into clinical use. Future research in this patient population should be targeted at: understanding physiological mechanisms underlying nutritional therapies, such as IM and probiotics, improving PEI diagnostics, and developing robust clinical trials that can assess the efficacy of these different nutritional strategies.

## Figures and Tables

**Figure 1 cancers-15-02466-f001:**
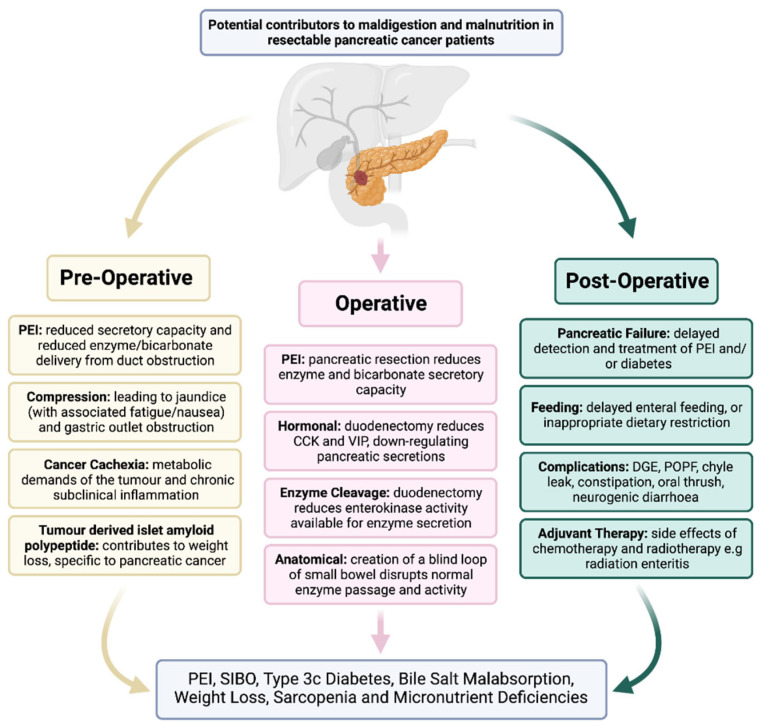
Causes of Malnutrition in Pancreatic Cancer Patients. Created with biorender.com.

**Figure 2 cancers-15-02466-f002:**
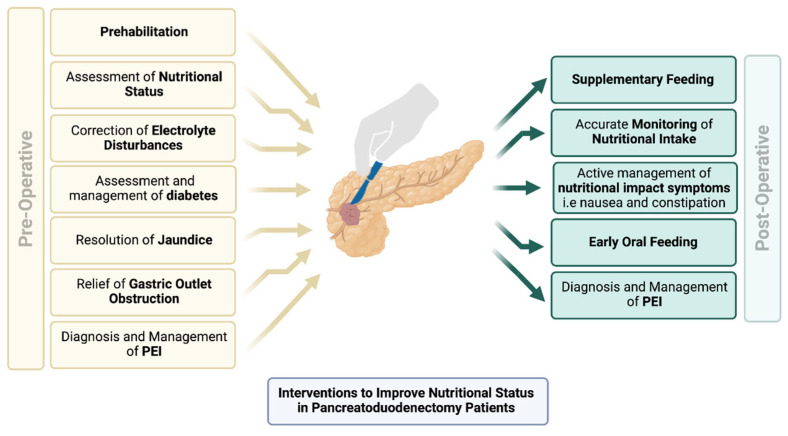
Pre- and post-operative interventions to improve the nutritional status of pancreatoduodenectomy patients. Created with biorender.com.

## Data Availability

Not applicable.

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
