# Peer review of "Recent Advances in Pancreatic Ductal Adenocarcinoma: Strategies to Optimise the Perioperative Nutritional Status in Pancreatoduodenectomy Patients"

_cancers, 2023, doi:10.3390/cancers15092466_

Round 1
Reviewer 1 Report
This review presents the current evidence for pre-, intra- and post-operative strategies to improve the nutritional status of patients undergoing Pancreaticoduodenectomy. And there are some suggestions which I’d like to point out:
1. This review elaborates in detail on the importance of preoperative nutritional assessment for PD patients, but after conducting nutritional assessment for patients, there is no in-depth discussion or suggestion on developing different preoperative nutrition support strategies and support durations for patients with different nutritional status.
2. Regarding which patients need to be placed with naso-jejunal (NJ) feeding tube during PD surgery, this review only mentioned that NJ feeding tubes should be placed for patients who are predicted with high risk of developing postoperative pancreatic fistula(POPF). However, according to the “Nutritional support and therapy in pancreatic surgery” guideline by ISGPS published in Surgery in 2018, it is also recommended to place NJ feeding tubes during pancreatic surgery for patients with severe preoperative malnutrition. Here I recommend that the authors could further review relevant researches and literatures and supplement this part. And we believe that it is reasonable to place of a feeding tube for enteral nutrition in patients undergoing PD when the patient presents with severe preoperative malnutrition.
3.it is mentioned that the risk and incidence of pancreatic exocrine insufficiency (PEI) may increase where radiotherapy has been administered as treatment for pancreatic cancer, either prior to, or following resection. Here I suggest authors could list some relevant research results or references to support this viewpoint.
Author Response
Thank you for taking the time to review this paper, and for your valuable comments, we hope we have addressed them all satisfactorily.
This review presents the current evidence for pre-, intra- and post-operative strategies to improve the nutritional status of patients undergoing Pancreaticoduodenectomy. And there are some suggestions which I'd like to point out:
1. This review elaborates in detail on the importance of preoperative nutritional assessment for PD patients, but after conducting nutritional assessment for patients, there is no in-depth discussion or suggestion on developing different preoperative nutrition support strategies and support durations for patients with different nutritional status.
Thank you, we have added a short sentence on individualizing strategies for pre-operative nutritional support but the in-depth variation in personalized strategies is beyond the scope of this review. We hope that we have covered the most important and controversial in the other sub-headings.
2. Regarding which patients need to be placed with naso-jejunal (NJ) feeding tube during PD surgery, this review only mentioned that NJ feeding tubes should be placed for patients who are predicted with high risk of developing postoperative pancreatic fistula(POPF). However, according to the "Nutritional support and therapy in pancreatic surgery" guideline by ISGPS published in Surgery in 2018, it is also recommended to place NJ feeding tubes during pancreatic surgery for patients with severe preoperative malnutrition. Here I recommend that the authors could further review relevant researches and literatures and supplement this part. And we believe that it is reasonable to place of a feeding tube for enteral nutrition in patients undergoing PD when the patient presents with severe preoperative malnutrition.
Thank you for your important comment. We agree and have now made this clearer in section 2.1, stating that NJ feeding should be used preoperatively in patients with severe malnutrition.
3. it is mentioned that the risk and incidence of pancreatic exocrine insufficiency (PEI) may increase where radiotherapy has been administered as treatment for pancreatic cancer, either prior to, or following resection. Here I suggest authors could list some relevant research results or references to support this viewpoint.
Thank you, we apologise for omitting these. We have added the relevant references to the manuscript.
Reviewer 2 Report
This is an interesting paper on a hot topic. I read it with interest and I have learnt something.
Just a couple of comment:
1) in the session 2.1 Preoperative Nutritional assessment I believe that you need to give some details about BIVA (bioelectrical impedance vector anlysis) and nutritional status in pancreatic cancer. there some interesting paper in the recent literature that you need to present here
2)line 61-62, I'm not sure your sentence is enough clear and understandable. What exactly do you mean with "..if patients are not certain requirements"?
3) in session 3 about Intraoperative strategies it would be suitable to give some details on fluids balance and restriction and strategies to reduce blood loss which is a definite rsik factor for complications and worse prognosis. I understand it could be argue that these are not strictly related to nutrition but I believe they matter in the general management of these complex patients.
Author Response
This is an interesting paper on a hot topic. I read it with interest, and I have learnt something. Just a couple of comment:
Thank you for taking the time to review this paper, and for your valuable comments, we hope we have addressed them all satisfactorily
In the session 2.1 Preoperative Nutritional assessment I believe that you need to give some details about BIVA (bioelectrical impedance vector analysis) and nutritional status in pancreatic cancer. there is some interesting paper in the recent literature that you need to present here
Thank you, we have added the below information about BIVA here and its relation to outcomes in cancer.
Bioimpedance vector analysis (BIVA) is a tool used to evaluate body composition (adipose and lean tissue, muscle and visceral fat tissue) and hydration status (total, intra and extracellular water). BIVA has been used to assess nutritional status in cancer patients and there is good evidence that this can be used to predict short- and long-term outcomes. Specific to pancreatic cancer BIVA has been shown to predict major morbidity following resection but further evidence is needed to define its role in routine pre-operative assessment.
line 61-62, I'm not sure your sentence is enough clear and understandable. What exactly do you mean with "..if patients are not certain requirements"?
Many thanks for highlighting this, corrected within the text. "if patients are not meeting certain requirements "
In session 3 about Intraoperative strategies it would be suitable to give some details on fluids balance and restriction and strategies to reduce blood loss which is a definite risk factor for complications and worse prognosis. I understand it could be argue that these are not strictly related to nutrition but I believe they matter in the general management of these complex patients.
Thank you for your important comment, we agree that this is an important point. We have therefore added a paragraph which details the harm relating from high levels of intraoperative blood loss and transfusion. We have also included details on strategies that can be used to reduce blood loss and transfusion.